# Splicing-Disrupting Mutations in Inherited Predisposition to Solid Pediatric Cancer

**DOI:** 10.3390/cancers14235967

**Published:** 2022-12-02

**Authors:** Piedad Alba-Pavón, Lide Alaña, Itziar Astigarraga, Olatz Villate

**Affiliations:** 1Pediatric Oncology Group, Biocruces Bizkaia Health Research Institute, 48903 Barakaldo, Spain; 2Pediatric Service, Hospital Universitario Cruces, 48903 Barakaldo, Spain; 3Pediatric Department, Universidad del País Vasco UPV/EHU, 48940 Leioa, Spain

**Keywords:** cancer predisposition syndromes, solid tumors, pediatric cancer, hereditary cancer, alternative splicing, mutations, genes

## Abstract

**Simple Summary:**

Until recently, the prevalence of hereditary cancer in children was estimated to be very low. However, recent studies suggest that at least 10% of pediatric cancer patients have a germline mutation in a cancer predisposition gene. It has been shown that most of these mutations affect splicing, a process by which different transcripts of the same gene are produced. The splicing process is very important, as it regulates many aspects of cellular proliferation, survival, and differentiation. Hereditary cancer genes are highly prone to splicing alterations, and among them there are several genes that may contribute to the development of pediatric solid tumors when mutated in the germline. In this review, we analyze the importance of the splicing-disrupting mutations in pediatric solid cancer and inherited predisposition syndromes. The therapies developed to correct aberrant splicing in cancer are also discussed.

**Abstract:**

The prevalence of hereditary cancer in children was estimated to be very low until recent studies suggested that at least 10% of pediatric cancer patients carry a germline mutation in a cancer predisposition gene. A significant proportion of pathogenic variants associated with an increased risk of hereditary cancer are variants affecting splicing. RNA splicing is an essential process involved in different cellular processes such as proliferation, survival, and differentiation, and alterations in this pathway have been implicated in many human cancers. Hereditary cancer genes are highly susceptible to splicing mutations, and among them there are several genes that may contribute to pediatric solid tumors when mutated in the germline. In this review, we have focused on the analysis of germline splicing-disrupting mutations found in pediatric solid tumors, as the discovery of pathogenic splice variants in pediatric cancer is a growing field for the development of personalized therapies. Therapies developed to correct aberrant splicing in cancer are also discussed as well as the options to improve the diagnostic yield based on the increase in the knowledge in splicing.

## 1. Alternative Splicing

Alternative splicing (AS) is a key mechanism that allows a single gene to increase its coding capacity, enabling the synthesis of distinct mRNA and protein [1]. AS determines many aspects of cellular proliferation, survival, and differentiation. Taking into account the importance of the splicing process in gene regulation, it is not surprising that alterations in this pathway have been implicated in several human cancers [2]. Analyses of more than 8000 tumors across 32 cancer types have revealed thousands of splicing variants not present in normal tissues, which are likely to generate cancer-specific markers and neoantigens [3,4]. The knowledge of the relationship between AS and epigenetic modifications has also enlarged the collection of biomarkers that can be used as cancer diagnostic and/or prognostic tools [5]. Moreover, aberrant splicing variants conferring drug or therapy resistance in tumors are more common than previously estimated [6].

The majority of studies on cancer and splicing have focused on the impact of somatic variants on alternative splicing events [3,7], but the association between splicing and germline variants in cancer predisposition genes is often overlooked. The discovery of pathogenic splice variants in pediatric cancer is a growing field that needs further investigation.

## 2. Cancer Predisposition Genes

Cancer predisposition genes are those in which germline mutations confer highly or moderately increased risks of developing neoplasms. The identification of these genes and the pathogenic variants found in them is essential for diagnosis and personalized treatment [8]. Thanks to the advances in next-generation sequencing (NGS), new cancer predisposition genes and pathogenic variants are being identified in pediatric tumors.

The prevalence of hereditary cancer in children was generally estimated to be very low until recent studies suggested that at least 10% of pediatric cancer patients carry a germline mutation in a cancer predisposition gene [9,10]. A significant proportion of the pathogenic variants associated with an increased risk of hereditary cancer are variants affecting splicing [11]. The identification of the variants that disrupt AS remains a challenge, and the consequence is that a significant proportion of patients with a possible hereditary cancer syndrome remain without a definitive molecular diagnosis. In a recent study of somatic mutations across 8656 tumor samples, the authors reported 1964 mutations that had originally been incorrectly classified and had clear evidence of creating alternative splice junctions [12].

It has recently been reported that hereditary cancer genes are highly susceptible to splicing mutations and that three main genes responsible for Lynch Syndrome, *MLH1*, *MSH2*, and *PMS2*, belong to a class of 86 disease genes that are enriched for splicing mutations [13]. It was also found that the COSMIC set of cancer genes [14] were overrepresented in these 86 splice-mutation-prone genes, with 20 of them being cancer-related genes (Table 1). This group of genes had a higher proportion of canonical splice sites and exonic mutations than the rest of the genes [13].

On the list of hereditary cancer genes that are highly susceptible to splicing mutations, there are several genes that may contribute to the development of pediatric solid tumors when mutated in the germline: *APC* [15,16], *ATM* [17], *BRCA1* [18,19], *BRCA2* [20], *FANCA* [15,21], *FANCD2* [19], *NF1* [22,23,24], *NF2* [20,25], *MLH1* [26], *MSH2* [26], *PMS2* [22,26], *RB1* [27,28,29,30], and *TSC2* [20] (Table 1).

Herein, we review the importance of ASin pediatric cancers, analyzing the germline splice variants described in genes that contribute to pediatric solid tumors and cancer predisposition syndromes. It is important to emphasize that more research is needed in this field since the identification of variants that affect splicing remains a challenge and most studies focus on consensus splice-site variants. Moreover, a better understanding of splicing biology will contribute toward the development of novel therapeutics for pediatric cancer.

To visualize the effects of the mutations that we are reporting in this review, we have represented the sequences that can be altered in Figure 1 and classified them into different groups according to the type of altered sequence: Type I, donor site region; Type II, acceptor site region; Type III, exonic region, including exonic splicing enhancers and silencers; and Type IV, intronic region, including intronic splicing enhancers and silencers.

## 3. Pediatric Solid Tumors

Solid tumors represent 60% of all pediatric malignant neoplasms, and the tumor types are very different from those found in adults. The most common pediatric tumors include central nervous system (CNS) tumors (35%); neuroblastoma (15%); soft tissue sarcoma (7%); Wilms tumor (6%); bone tumors, including osteosarcoma and Ewing sarcoma (8%); retinoblastoma (5%); and other rare tumors, including hepatoblastoma, germ cell tumors, and melanoma (17%) [31]. In this review, we focus on the most prevalent tumors in childhood: CNS tumors, sarcomas, and blastomas (neuroblastoma, retinoblastoma, and Wilms tumor).

### 3.1. CNS Tumors

CNS tumors are the second most common type of cancer among children, and they often occur in patients with a cancer predisposition syndrome [21]. The following subtypes are discussed:

#### 3.1.1. Medulloblastoma

Medulloblastoma (MB) is the most common malignant brain tumor in children [32], and recently the World Health Organization (WHO 2021) classified MB at the molecular level into four different types: MB WNT-activated, SHH-activated, group 3, and group 4 [33,34,35]. MBs arise in the cerebellar vermis and spread rapidly through the cerebrospinal pathways [36]. AS is especially prevalent in the mammalian nervous system, including the cerebellum, where it modulates relevant processes (neural tube patterning, synaptogenesis, membrane physiology, and synaptic plasticity), so a disruption of splicing regulation can promote pathogenic events [37].

Menghi et al. investigated patterns of differential splicing between pediatric MBs and the normal cerebellum on a genome-wide scale and concluded that inappropriate splicing frequently occurs in human MBs and may be linked to the activation of developmental signaling pathways and a failure of cerebellar precursor cells to differentiate [7]. Moreover, splicing patterns are distinct and specific between molecular subgroups [38]. Subgroup-specific splicing and alternative promoter usage were most prevalent in group 3 and SHH MBs, while they were less frequent in WNT and group 4. AS events in MB may be partially regulated by the correlative expression of antisense transcripts, suggesting a mechanism affecting subgroup-specific AS [38].

Suzuki et al. discovered that approximately 50% of SHH MBs harbor a somatic mutation in the 5′ splice-site binding region of U1 spliceosomal small nuclear RNAs (snRNAs). This mutation is not present across other MB subgroups. SnRNA mutant tumors have significantly disrupted AS, and as a result aberrant AS inactivates *PTCH1* and activates oncogenes (*GLI2* and *CCND2*), representing a novel target for therapy [39].

MB tumors may appear sporadically or as a part of an inherited syndrome. Pathogenic germline mutations in known cancer predisposition genes have an important role, mainly in WNT-activated and SHH-activated MB [15]. In a recent study, germline data in 1022 patients with MB were analyzed, and the results showed a significant excess of pathogenic mutations in the *APC*, *BRCA2*, *PALB2*, *PTCH1*, *SUFU*, and *TP53* genes [15]. Splice variants in the canonical sites and splice regions were found in the *ATM*, *BRCA2*, *FANCA*, *FANCC*, *PALB2*, *PTCH1*, *RAD51C*, *SUFU*, *WRN*, *WT1*, and *XPC* genes (Table 2).

For SHH-activated MB, the Gorlin (*PTCH1* and *SUFU*) and the Li–Fraumeni syndromes (*TP53*) are the most common predisposition syndromes [56,57,58,59]. Additional candidates for SHH-activated MB include *BRCA2* and *PALB2*, which can be associated to Fanconi anemia [15,60,61]. About 5% of patients with Gorlin syndrome (GS) develop MB, mainly the desmoplastic form [62]. Between 50% and 85% of patients with GS have germline mutations in *PTCH1*. In a study of GS, *PTCH1* was analyzed in two familial and three sporadic GS cases, and five germline mutations were found in *PTCH1* [42]. One of them was a splice-site mutation (c.584+2T>G) in an 11-year-old male patient who developed MB at the age of 1 year (Table 2).

*SUFU* is also involved in the susceptibility to MB. In a report, they identified the c.1022+1G>A *SUFU* germline splice mutation in a family that was *PTCH1*-negative but had signs and symptoms of GS, including MB [43]. Another study described a family previously diagnosed with GS with a novel *SUFU* splice-site pathogenic variant (c. 1365+2T>A) [44]. Germline *SUFU* mutations were analyzed in children with desmoplastic/nodular MB, and eight germline mutations were found, with three of them being splice variants (c.182+3A>T; c.318-10delT; and c.1297-1G>C) [45]. Another report showed *SUFU* germline mutations in desmoplastic MBs, one of them located in the conserved splice acceptor site of exon 2 (Table 2) [46].

Li–Fraumeni syndrome (LFS) is a rare autosomal dominant form of familial cancer, characterized by the early onset of diverse malignancies, including sarcomas, brain tumors, and leukemias [63]. Germline mutations in *TP53* have primarily been identified in LFS [64]. Mutations in splice sites are also very frequent in LFS, while missense mutations are less common in comparison to other familial or sporadic cancers [65]. Several studies have described splice-site mutations in LFS [55,66,67], but we have only found one study in the IARC *TP53* database, which described one splice variant in *TP53* in an LFS pediatric patient with MB (c.376-2A>G) [47]. Splice variants in *TP53* have been found in a pediatric patient with choroid plexus carcinoma (c.560-2A>C) [55] and in a pediatric glioblastoma patient (c.919+1G>A) (Table 2) [68].

Recently, germline splice mutations in other genes such as *ELP1* have been found in two independent families with SHH-activated MB [40]. For WNT-activated MB, Turcot Syndrome (*APC*) is the most common predisposition syndrome [56], a rare disorder characterized by the association of colonic polyposis and primary brain tumors [69]. In MB associated with *APC* germline pathogenic variants, no splice variants were found in this review.

Another less common MB-associated syndrome is ataxia telangiectasia (AT) [70]. AT is an autosomal recessive disease characterized by neurological and immunological symptoms, radiosensitivity, and cancer predisposition. The mutated gene in AT is *ATM*, and different splice variants of this gene have been described in pediatric MB (Table 2) [15]. Moreover, a germline splice variant of the *MUTYH* gene has been described in a pediatric patient with MB (Table 2) [22].

#### 3.1.2. Gliomas

Gliomas are CNS neoplasms that affect both the brain and spinal cord, and they are the most common primary CNS tumors, mainly astrocytomas [71].

##### Low-Grade Gliomas (LGGs)

LGGs and glioneural tumors represent over 30% of pediatric CNS neoplasms [72,73]. Within the LGG category, there are different tumor types and subtypes:a. *Astrocytoma*

Pilocytic astrocytoma is the most common type in children and young adults [72]. In relation to genetic predisposition, one study showed germline splice mutations in the *MUTYH* and *ERCC2* genes [51] in a highly infiltrative astrocytoma and a diffuse astrocytoma, respectively, although the variants were not annotated in the manuscript (Table 2). In a recent study, a novel *CHEK2* splice variant (c.444+1G>A) was identified in a 7-year-old child diagnosed with a subependymal giant cell astrocytoma (Table 2) [48]. Functional studies have shown the use of an alternative 5′ splice site that creates a premature stop codon. As a result of this change, the transcript is truncated, which results in reduced CHEK2 protein levels [74].

b. 
*Ependymoma*


Ependymoma (EP) is the second most common malignant brain tumor in children, and it originates from the walls of the ventricular system [75]. The etiology is largely unknown, and germline DNA sequencing studies on pediatric EP are scarce. Pathogenic germline variants in known cancer predisposition genes have been detected in genes such as *NF2*, *LZTR1*, *NF1*, and *TP53* [76]. EP can be associated with type 2 neurofibromatosis with a high proportion of pathogenic mutations in *NF2*. The most common alterations in *NF2* are splice-site or nonsense mutations, but these are mostly found in intracranial meningiomas and other adult nervous system cases [77,78,79], except the variant c.447+1G>A, which was described in a pediatric EP (Table 2) [22].

Epigenetic alterations appear to play a central role in the development of the molecular classification of EPs [80]. Recent findings have shown that posterior fossa type A (PFA) EPs exhibit low H3K27 methylation and overexpress EZHIP (enhancer of zeste homologs inhibitory protein), which dysregulates gene silencing to promote tumorigenesis. Genomic dataset analyses from PFA and diffuse intrinsic pontine gliomas (DIPG) have revealed that these two different tumors share a common dysregulated chromatin landscape [81].

c. 
*Optic glioma*


Neurofibromatosis type 1 (NF1) is one of the most frequent autosomal dominant disorders and is caused by mutations in the *NF1* gene. NF1 patients are predisposed to develop brain tumors, among others, and gliomas are found in 15–20% of affected individuals [82,83]. About 15% of children with NF1 develop low-grade optic pathway gliomas (OPG) [84], whereas high-grade gliomas, including anaplastic astrocytomas (AA) and glioblastomas, are less frequent in children with NF1 [23,85].

In one study, NF1 patients were analyzed (31% with OPG), and an *NF1*-splice germline variant was found in a OPG patient: c.2325+1G>A, which produced exon 14 skipping (Table 2) [50]. Different *NF1* germline mutations in pediatric glioma patients that affected the splicing process have been described: c.205_205insTC, c.1185+1G>A, c.889-2A>G, c.2325+1G>A, and c.1260+1G>T in pilocytic astrocytomas and OPG [49]. Moreover, the variant c.6641+1G>A has been found in a pediatric LGG (Table 2) [22].

##### High-Grade Gliomas (HGGs)

HGG is one of the most fatal childhood brain tumors and can be associated with underlying cancer predisposition syndromes such as NF1 and Turcot and Li–Fraumeni syndromes [86].

*NF1* germline mutations have been described in high glioma pediatric patients, affecting the splicing process as c.1641+2T>A and c.4174-2>AG in glioblastoma and anaplastic astrocytoma (AA), respectively (Table 2) [49].

Constitutional mismatch repair deficiency (CMMRD) is a syndrome caused by biallelic mutations in the mismatch repair pathway [87]. This repair system comprises different genes, including *MSH2*, *MSH6*, *MLH1*, and *PMS2* [88]. Patients with CMMRD or familial adenomatous polyposis (FAP) who develop brain tumors were lumped together under the term Turcot syndrome [16]. Biallelic germline splice mutations in *MSH6* have been reported in MB and in glioblastoma multiforme (Table 2) [41]. In a report, an inactivating germline mutation in *MUTYH* was found in a patient with a high-grade midline glioma (Table 2) [52]. A germline mutation in *ATM*, affecting the splicing process, was found in a pediatric HGG patient (Table 2) [22].

#### 3.1.3. Other CNS Tumors

##### Pinealoblastoma

The DICER1 syndrome is related to several benign and malignant tumors, including rhabdomyosarcoma and pinealoblastoma [89]. A germline *DICER1* splice-site variant (c.4050+1G>A) was found in a 10-year-old patient with pinealoblastoma (Table 2) [54].

##### Atypical Teratoid/Rhabdoid Tumors

Rhabdoid tumors (RTs) are most commonly observed in the brain, where they are called atypical teratoid/rhabdoid tumors (AT/RT) [53]. The majority of RTs are caused by a loss of function in *SMARCB1*, and more recently mutations in *SMARCB4* have been found as a cause of RTs. Germline mutations in *SMARCB1* are also associated with familial schwannomatosis [90]. Deletions or truncating mutations of *SMARCB1* are generally found in AT/RT, and loss-of-function mutations in exon 1 and splice-site mutations are more frequent in schwannomatosis [90]. Kordes et al. analyzed 50 patients with AR/RT, and germline mutations in *SMARCB1* were detected in 10 patients, including one splice-site mutation: c.501-2A>G (Table 2) [53].

In another report, they showed an inherited *SMARCB1* mutation in a two-generation family that was a splice-site mutation in exon 7 [91]. In a recent study, they presented two siblings with congenital AT/RT due to a germline SVA-E retrotransposon insertion into intron 2 that disrupts the splicing between exons 2 and 3 of *SMARCB1* [92].

### 3.2. Sarcomas

Sarcomas are tumors with a mesenchymal origin that comprise around 12% of all neoplasms in children and adolescents. They are a very heterogeneous group of tumors, comprising more than 70 distinct histological subtypes. Sarcomasare classified into two main groups: bone and soft-tissue sarcomas, with osteosarcoma, Ewing sarcoma, and rhabdomyosarcoma being the most frequent types in children and adolescents [93].

#### 3.2.1. Osteosarcoma

Osteosarcoma is the most common primary bone tumor. The peak incidence occurs during the pubertal growth spurt [94]. It was estimated that 10% of osteosarcoma patients have a hereditary predisposition syndrome [95]. However, recent publications estimated that 28% of patients diagnosed with osteosarcoma had pathogenic/likely pathogenic germline variants [18]. Many different cancer predisposition syndromes are associated with osteosarcoma development, including autosomal dominant disorders (LFS and hereditary retinoblastoma [96,97]) and autosomal recessive disorders (primarily DNA helicase disorders: Rothmund–Thomson, RAPADILINO, Werner, and Bloom syndromes [98,99,100]).

LFS is associated with germline loss-of-function mutations in *TP53*. Several germline variants have been described in *TP53* affecting mRNA splicing, some of them associated with osteosarcoma development in pediatric patients (Table 3). A rare *TP53* germline mutation, c.671+1G>A, was described in a 15-year-old patient diagnosed with osteosarcoma in an LFS context. This variant results in a 6-amino-acid insertion between codons 224 and 225 in exon 6 [101]. The *TP53* c.672G>A germline variant was reported in a 17-year-old male with two primary sarcomas (pleomorphic sarcoma and telangiectatic osteosarcoma) (Table 3). This variant is a synonymous change that preserves the glutamate in position 224, but the change results in a shift of the exon 6 splice site by five base pairs, producing a frameshift and a premature stop codon at residue 246 in exon 7 [102].

Hereditary retinoblastoma is an autosomal dominant syndrome that is linked to *RB1* germline mutations. The primary tumor developed in childhood is retinoblastoma; however, there is an increased risk of developing various neoplasms, especially osteosarcoma [116]. Atypical *RB1* germline variants have been described in sarcoma patients without retinoblastoma as a primary tumor [117]. Two pathogenic splice variants have been found in the germline in two different patients diagnosed with osteosarcoma (Table 3) [22,107].

Syndromes characterized by germline mutations in genes encoding DNA helicases of the RecQ family have an increased risk for cancer, especially osteosarcoma. These are Rothmund–Thomson, RAPADILINO, Werner, and Bloom syndromes [118]. Rothmund–Thomson and RAPALIDINO syndromes are caused by mutations in the *RECQL4* gene. Rothmund–Thomson syndrome is a disorder characterized by poikilodermatous skin changes, congenital skeletal abnormalities, premature aging, and an increased risk for cancer [119]. RAPALIDINO is a very rare syndrome identified by radial hypoplasia, patellae hypoplasia, a cleft or highly arched palate, diarrhea, dislocated joints, small size and limb malformation, a slender nose, and normal intelligence. In the Human Gene Mutation Database (HGMD), 14% of the reported variants in the *RECQL4* gene are splice variants, and 7 of the 25 splice variants are described in patients diagnosed with Rothmund–Thomson syndrome who developed osteosarcoma (Table 3) [108,109,110,111].

Werner syndrome is a disease caused by mutations in the *WRN* gene, and it is associated with the development of osteosarcoma during adult life [95]. Bloom syndrome is caused by mutations in another DNA helicase gene, *BLM*. This syndrome is characterized by clinical features including small stature, photosensitive rashes, and immunodeficiency [95]. Ten percent of *BLM* variants reported in the HGMD are splice variants; however, none of these variants are associated with the development of osteosarcoma.

#### 3.2.2. Ewing Sarcoma

Ewing sarcoma is the second most common bone and soft tissue cancer. The majority of Ewing sarcomas arise in bone, and up to 30% arise in soft tissue. The highest incidence is in the second decade of life. Ewing sarcoma development is uncommon in patients younger than 5 years or older than 30 years [120,121,122].

Ewing sarcoma is characterized by a low somatic mutation rate, and it is mainly caused by a chromosome rearrangement as a driver alteration. This rearrangement is between the *EWSR1* gene and members of the ETS gene family. The most common is the EWSR1-FLI1 fusion gene [123]. Most of studies related to Ewing sarcoma predisposition have focused on the identification of susceptibility loci from genome-wide association studies (GWASs) [124].

Aberrant splicing of the EWS-FLI1 transcript alters EWS-FLI1 protein expression and EWS-FLI1-driven expression [125]. Targeting EWS-FLI1 is one of the therapeutic options, but recently epigenetic/transcriptional modulators have been proven to be promising therapeutic strategies for indirectly altering its expression and/or function [126].

EWS-FLI1 induces the expression of a specific set of novel spliced and polyadenylated transcripts in regions of the genome that are normally transcriptionally silent. These neogenes are practically undetectable in normal tissues or non-Ewing-sarcoma tumors [127].

Recently, germline pathogenic variants have been described in genes involved in DNA damage repair in Ewing sarcoma patients [19,107]. In these studies, germline splice variants have been found in the *NTHL1*, *SLX4*, *CHEK2*, *EXT2*, *RAD51C*, *FANCA*, *FANCC*, and *FANCD2* genes (Table 2), most of them described in splice sites. All of these genes, except *EXT2*, are involved in the DNA repair response through different signaling pathways such as the nucleotide-excision repair (NER) pathway, DNA double-strand break repair, the DNA damage checkpoint, and oxidative DNA damage repair [128,129,130,131,132,133,134,135]. A loss of the functionality of DNA repair proteins could contribute to rearrangement signatures due to the failure of homologous recombination mechanisms [19,107]. Germline mutations in some of these genes are associated with Fanconi anemia (*SLX4*, *FANCA*, *FANCC*, and *FANCD2*) [136,137].

#### 3.2.3. Rhabdomyosarcoma

Rhabdomyosarcoma (RMS) is the most common soft tissue sarcoma developed at a pediatric age. The two main subtypes are embryonal and alveolar RMS [93,138]. Chromosomal translocations involving chromosomes 1 or 2 and chromosome 13 are associated with 80% of alveolar RMS cases. These rearrangements fuse the *PAX3* or *PAX7* and *FOXO1* genes [139]. Embryonal is the most common subtype, and though translocations are not observed, *TP53*, *KRAS*, *NRAS*, *HRAS*, *CTNNB1*, and *FGFR4* are the most frequently mutated genes in this subtype [140].

Most rhabdomyosarcomas are primarily sporadic, but they can be associated with several syndromes, including RASopathies and Li–Fraumeni and DICER1 syndromes [141,142,143]. Cancer predisposition syndromes are more frequent in patients with embryonal RMS than in those with the alveolar subtype [138].

RASopathies are a group of disorders caused by germline mutations in the genes involved in the RAS/MAPK signaling pathway with a high risk of cancer development [144,145]. NF1, Costello syndrome, and Noonan syndrome are the RASopathies most frequently associated with the risk of RMS development [146,147]. Costello syndrome is caused by *HRAS* germline mutations. Noonan syndrome is associated with mutations in the RAS family of genes, *PTPN11*, and *SOS1* genes. The RAS family are GTPases that catalyze the hydrolysis of GTP, activating the MAPK signaling pathway. They are oncogenes that exhibit activating mutations in cancer. Germline splice variants have not been described in the *KRAS*, *HRAS*, and *NRAS* genes in the HGMD database. In contrast, many splice variants have been described in the *SOS1* and *PTPN11* genes, but none of them were in patients with RMS.

The *DICER1* gene encodes an enzyme involved in the production of mature microRNAs [148]. *DICER1* germline mutations cause a cancer predisposition syndrome with cancer risk for pleuropulmonary blastoma, cystic nephroma, Sertoli–Leydig cell tumors, pinealoblastoma, and embryonal RMS [149]. In this context, the *DICER1* c.1907+1G>A splice variant was found in the germline in a 6-month-old female with an embryonal rhabdomyosarcoma localized in the vagina (Table 3) [115].

Many of the previously mentioned cancer predisposition syndromes are also associated with the development of rhabdomyosarcomas, in particular LFS and NF1. Different splice variants in genes associated with these syndromes, *TP53* and *NF1*, are described in RMS pediatric patients in the germline, all of them described in splice sites (Table 3).

### 3.3. Neuroblastoma

Neuroblastoma (NB) originates from neural crest cells and affects the sympathetic nervous system. It is characterized by an early age of onset and a high frequency of metastatic disease at diagnosis in patients over 1 year of age. NB tumors present few chromosomic aberrations, including MYCN amplification, 17q gain, 1p deletion, and 11q deletion [150].

NB has been associated with the following cancer predisposition syndromes: familial neuroblastoma, familial paraganglioma/pheochromocytoma, CCHS/Hirschsprung, Beckwith–Wiedemann, Simpson–Golabi–Behmel, LFS, Sotos, Costello, Noonan, Rubinstein–Taybi, Wolf–Hirschhorn, Weaver, NF1, ROHHAD, and Fanconi anemia [136,151,152,153,154,155,156,157,158,159,160,161,162,163,164,165,166,167,168].

The *PHOX2B* and *ALK* genes are major susceptibility genes of familial NB [150]. *PHOX2B* encodes a transcription factor promoting neural crest differentiation. NB-exclusive mutations are mainly missense and frameshift; splicing variants of this gene have not yet been associated with NB [169]. *ALK* was also identified as major familial neuroblastoma predisposition gene [170], and as in the previous case, no NB-associated splicing variants have been described.

For patients with sporadic disease, different studies focused on uncommon germline variants associated with NB have been conducted, and pathogenic and likely pathogenic variants were identified in predisposition genes such as *ALK*, *CHEK2*, *BRCA2*, *SMARCA4*, and *TP53* and in candidate genes such as *AXIN2*, *PALB2*, *BARD1*, *PINK1*, *APC*, *BRCA1*, *SDHB*, and *LZTR1* [20,22,171,172,173,174]. A pathogenic germline splice variant in *BRCA2*, c.8488-1G>A, was identified to be associated with NB [22]. *PALB2* also had a germline variant that was predicted to delete a splice donor site, c.1684+1C>A [172].

### 3.4. Retinoblastoma

Retinoblastoma (RB) is the most common primary malignant intraocular cancer in children, and it represents 3% of all pediatric tumors [175]. There are different forms of RB: unilateral or unifocal, bilateral or multifocal, and trilateral [175].

Overall, around 90% of bilateral cases and 10–25% of unilateral cases have *RB1* germline mutations [176]. *RB1* is a tumor-suppressor gene and encodes pRB, a key regulator of the cell cycle [143]. *RB1* is one of the hereditary cancer genes that is highly susceptible to splicing mutations. In fact, aberrant splicing of the *RB1* gene was found to be the dominant cause of retinoblastomas in a recent study [177]. In this report, they observed that, of all the diseases collected in the HGMD, the highest proportion of splicing phenotypes seen in exonic mutations was found in *RB1*. These data suggested that *RB1* is particularly susceptible to splicing mutations [177]. Consistent with the above, germline mutations affecting *RB1* alternative splicing have been identified in many studies of RB patients [22,176,177,178,179,180,181,182].

### 3.5. Wilms Tumors

Wilms tumor (nephroblastoma, WT) is the most common pediatric renal malignancy, representing 90% of renal tumors and 5–7% of all pediatric malignancies [175]. It is estimated that about 10% of WT cases are caused by germline pathogenic variants or epigenetic alterations occurring early during embryogenesis [183]. WT is primarily a nonhereditary condition [184].

WT is associated with different hereditary cancer syndromes, including WAGR, Denys–Drash, Bloom, Frasier, Gorlin, Beckwith–Wiedemann, Sotos, Simpson–Golabi–Behmel, Perlman, mosaic variegated aneuploidy, Muliebry nanism, hereditary hyperparathyroidism, isolated hemihypertrophy, LFS, DICER1, and Bohring–Opitz syndromes among others [184,185,186,187,188,189,190,191,192,193,194,195,196,197,198,199,200,201,202,203,204,205,206,207].

There are more than 20 WT predisposition genes. There is an overlap of only four genes, *WT1*, *IGF2*, *TP53*, and *DICER1*, between the WT predisposition genes and the somatically mutated WT driver genes [183,208].

Regarding the splicing-disrupting mutations, the variant c.1095G>T in *CHEK2* was shown to affect AS. This variant increases the expression of a transcript without exon 10, which loses the kinase function of the protein [208].

One of the genetic syndromes associated with WT, Frasier syndrome (FS), is caused by splicing variants that affect the balance of *WT1* isoforms. Two alternative splice donor sites in intron 9 are responsible for creating two different transcripts (with or without lysine-threonine-serine), and an imbalance in the transcripts results in the development of FS. Pathogenic variants in this intron have been identified in WT patients [209].

An interesting case report described a pediatric patient with no response to treatment to a bilateral WT carrying a novel germline *WT1* gene splice-site mutation in intron 6, c.895-2A>G. The authors suggested that the correlation of this variant with response and prognosis should be further studied [210].

A germline mutation affecting splicing in the *CTR9* gene has been identified in a family with WT [211]. The variant c.958-2A>G produces exon 9 skipping, and it is predicted to encode a truncated protein. Another splice variant was found in *CTR9*, the splice-site mutation c.1194+2T>C, which is predicted to disrupt the exon 9 splice site, which was analyzed and confirmed with a minigene strategy [212]. A pathogenic germline splice variant has been also found in the *TRIM28* gene, a WT predisposition gene: c.840–2A>G [213].

## 4. Therapeutic Targeting of Splicing in Cancer

The identification of cancer-specific splice variants has increased the development of new therapies to correct aberrant splicing. Different strategies have been used for this purpose, such as blocking components of the spliceosome, targeting protein isoforms produced by incorrect AS, blocking protein kinases that regulate splicing factors, and the use of antisense oligonucleotides (ASOs), among others [214].

Small molecules that are modulators of the spliceosome have been tested in cancer clinical trials, for example, modulating the splicing factor SF3B [215]. Synthetic analogues of compounds derived from bacteria that are cytotoxic to cancer cell lines were designed to bind SF3B [215]. Upon binding to the splicing factor, they prevent the assembly of the spliceosome, thus inhibiting splicing [216]. Changes in splicing are mainly in genes related to cell cycle regulation and apoptosis [215].

The application of ASOs in cancer therapy is still under intense research, but promising preclinical results have been reported [215]. Results obtained in clinical trials are also encouraging [217]. ASOs can correct cancer-related AS, as it has been shown in cancer cell lines (Figure 2) [215,218]. For all these reasons, there is a growing interest in the use of ASO-based therapeutics in cancer [6,219]. ASOs are particularly interesting in cancer therapy, as they can be generated for specific target sequences. They can decrease the expression of coding oncogenic drivers, and they can target noncoding RNAs. Nevertheless, ASOs have not yet obtained marketing authorization for cancer treatment [218].

There are several challenges that may impact the therapeutic efficacy of oligonucleotide therapeutics in cancer [217]. One of them is to achieve the efficient delivery of the drugs to cancer cells in the body. Oligonucleotides need to overcome several barriers, such as the vascular endothelial barrier or the blood–brain barrier, depending on the target tissue and avoid rapid clearance from circulation to obtain a therapeutic effect [220]. The blood–brain barrier seems to be impervious to oligonucleotides. Many attempts have been performed to deliver oligonucleotides across this barrier, with modest success [220]. The most promising involve conjugates of oligonucleotides with cell-penetrating peptides [221], but there are concerns about the possible toxicities of the peptides.

Other limitations for antisense therapeutics are the complexity of cancers that sometimes involve multiple genes to target and drug interactions. It has been reported that oligonucleotides may compete with chemotherapeutics for plasma protein binding, which can reduce the in vivo efficacy of the combination compared to chemotherapeutics alone [222].

**Figure 2 cancers-14-05967-f002:**
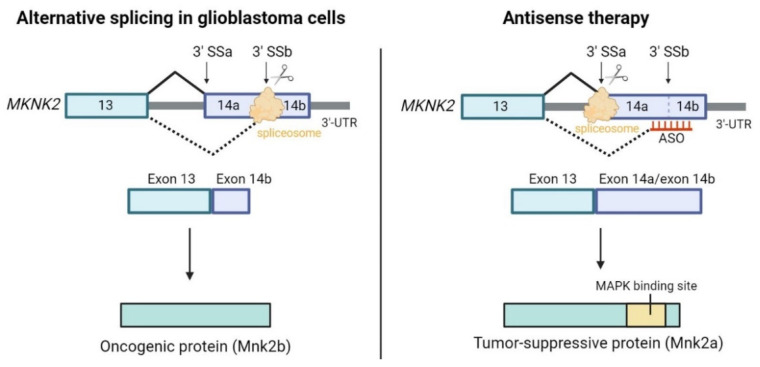
Therapeutic strategy using antisense oligonucleotides. *MKNK2* encodes the kinase Mnk2, and exon 14 is defined by two different alternative 3′ splice sites (3′SSa and 3′SSb). In glioblastoma cells, the spliceosome binds to the 3′SSb splice site, producing the oncogenic isoform Mnk2b. Using an ASO to block this site favors the use of 3′SSa, and the tumor-suppressive protein Mnk2a is produced [223].

Regarding cancer pharmacogenetics, somatic mutations have become druggable targets or biomarkers, whereas germline mutations are potentially responsible for drug responses [224]. Primary resistance to treatments can be supported by germline splicing variants. An example of this is the tyrosine kinase inhibitor (TKI) imatinib and BIM-γ [225]. There is a recurrent deletion in intron 2 of the *BIM* gene that was found to be associated with an increased likelihood of chronic myeloid leukemia resistance to imatinib and second-line TKIs [225].

Research on splicing-disrupting mutations in inherited predispositions to solid pediatric cancer and clinical trials are necessary. At this time, there are no clinical trials registered at ClinicalTrial.gov on splicing that include pediatric patients with solid tumors (Figure 3). At the time of this review, there are 1193 registered clinical trials in pediatric solid tumors, but none of them are related to therapies to correct aberrant splicing.

At the time of this review there are no active registered trials at ClinicalTrial.gov using ASOs for pediatric cancer in general. There was a clinical trial that included pediatric patients, which was already completed, to treat patients with advanced melanoma using Bcl-2 ASOs in combination with dacarbazine (NCT00016263). However, for adult cancer there are 18 active trials registered with ASO technology, mainly for lymphomas, leukemias, and solid tumors.

## 5. Conclusions and Future Perspectives

In this review, we have shown that most of the splicing variants described in the germline in pediatric solid tumors are located in the consensus splice sites. The identification of variants that affect splicing remains a challenge, and most studies only focus on consensus splice-site variants. As a result, variants in exons or deep introns are not studied, and many patients remain undiagnosed. Moreover, it is important to experimentally verify the impact of splicing on variants outside the canonical splice sites to ensure the accurate classification of variants. In this regard, the identification and study of variants for which *in silico* analyses predict an unknown significance but could alter the splicing process would provide new insights into cancer pathogenesis.

To increase the diagnostic rate, RNA sequencing (RNA-seq) has a great potential for improving diagnosis because of the splicing results generated by this analysis [226]. RNA-seq provides an opportunity to identify pathogenic variants in the noncoding regions of genes [227]. Several reports have also shown the benefits of RNA-seq for hereditary cancer predisposition genes. In a recent study, RNA analyses allowed the classification of 88% of the cancer gene splicing variants selected for analysis as either pathogenic or benign. These studies show that patients under DNA analysis would benefit from the addition of RNA-seq to the diagnosis [228].

The additional increase in diagnostic yield offered by RNA-seq represents an opportunity for the development of new personalized management strategies that could contribute to improving early detection, therapy, and prognosis [229]. Identifying a cancer predisposition syndrome has a huge impact in the clinical management of pediatric cancer patients and their families, allowing a better follow-up and adequate genetic family counselling.

## Figures and Tables

**Figure 1 cancers-14-05967-f001:**
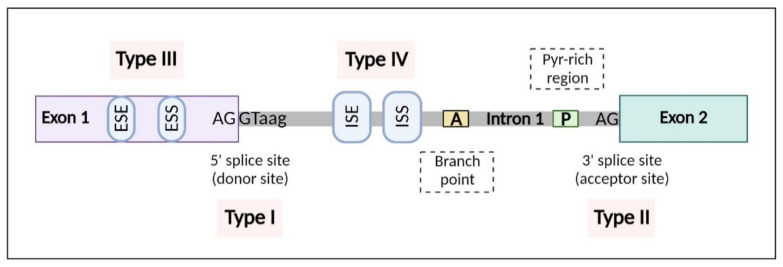
Schematic representation of the splicing sequences that can be altered by mutations and their classification for this review. The DNA sequences include donor and acceptor splice sites (Type I and Type II, respectively); exonic sequences, including exonic splicing silencers (ESS) and enhancers (ESE) (Type III); and intronic sequences, including intronic splicing enhancers (ISE) and silencers (ISS) (Type IV).

**Figure 3 cancers-14-05967-f003:**
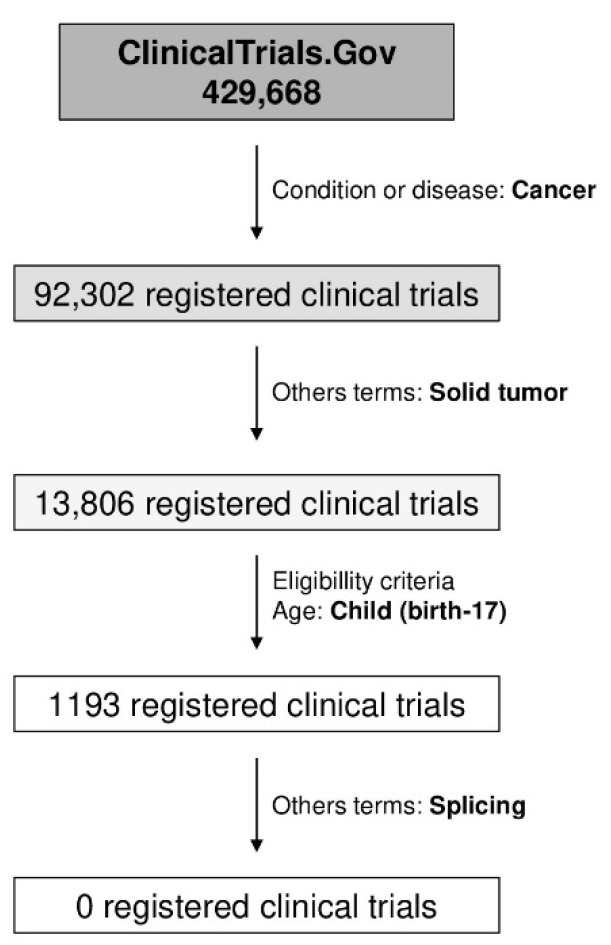
Clinical trials that include research on splicing in pediatric cancer. There are 429,668 registered clinical trials in the ClinicalTrial.gov database at the time of this review. The number is minimized to 13,806 when filtering the trials by the terms cancer and solid tumor. Taking into account the age (child: birth–17 years), there are 1193 clinical trials. When the term splicing is included, no clinical trials are registered.

**Table 1 cancers-14-05967-t001:** Cancer-related genes enriched in splicing alterations.

Gene ^a^	Pathway/Function	Associated Cancer Predisposition Syndrome
*APC*	Tumor suppressor	Familial adenomatous polyposis
*ATM*	Tumor suppressor	Ataxia telangiectasia
*BRCA1*	Tumor suppressor	Hereditary breast and ovarian cancer syndrome
*BRCA2*	Tumor suppressor	Hereditary breast and ovarian cancer syndrome
*COL1A1*	Pro-alpha1 chains of type I collagen	-
*COL2A1*	Pro-alpha1 chains of type II collagen	-
*ELN*	Elastic fiber formation	-
*EXT1*	Tumor suppressor	Hereditary multiple exostoses, Langer–Giedion syndrome
*FANCA*	Fanconi anemia complementation group A	Fanconi anemia
*FANCD2*	Fanconi anemia complementation group D2	Fanconi anemia
*FANCG*	Fanconi anemia complementation Group G	Fanconi anemia
*MLH1*	Tumor suppressor	Lynch syndrome
*MSH2*	Tumor suppressor	Lynch syndrome
*NF1*	Tumor suppressor	Neurofibromatosis type 1
*NF2*	Tumor suppressor	Neurofibromatosis type 2
*PMS2*	Tumor suppressor	Lynch syndrome
*PRKAR1A*	Protein Kinase CAMP-Dependent Type I Regulatory Subunit Alpha/tumor suppressor	Carney complex
*RB1*	Tumor suppressor	Retinoblastoma
*TSC2*	Tumor suppressor	Tuberous sclerosis type 2
*WAS*	Effector protein for Rho-type GTPases	Wiskott–Aldrich syndrome gene

^a^ Genes predisposing to solid pediatric tumors are highlighted in grey.

**Table 2 cancers-14-05967-t002:** Germline pathogenic splice variants found in pediatric patients with CNS tumors.

Gene	Diagnosis	Variant	Type of Mutation	Reference
*ATM*	Medulloblastoma	c.6095G>A	I	[15]
*ATM*	Medulloblastoma	c.2921+1G>C	I	[15]
*BRCA2*	Medulloblastoma	c.631+2T>G	I	[15]
*BRCA2*	Medulloblastoma	c.-39-1_-39delGA	II	[15]
*ELP1*	Medulloblastoma	c.3700+1G>A	I	[40]
*ELP1*	Medulloblastoma	c.3572+1G>A	I	[40]
*ELP1*	Medulloblastoma	c.2959-1G>T	II	[40]
*ELP1*	Medulloblastoma	c.741-1G>T	II	[40]
*ELP1*	Medulloblastoma	c.649G>A	I	[40]
*ELP1*	Medulloblastoma	c.2959-1G>T	II	[40]
*FANCA*	Medulloblastoma	c.2778+1G>A	I	[15]
*FANCC*	Medulloblastoma	c.996+1G>T	I	[15]
*MSH6*	Medulloblastoma	c.(4002-31_4002-8delins24) + (4002-31_4002-8delins24)	IV	[41]
*MUTYH*	Medulloblastoma	c.925-2A>G	II	[22]
*PALB2*	Medulloblastoma	c.3201+1G>C	I	[15]
*PTCH1*	Medulloblastoma	c.1729-2A>G	II	[15]
*PTCH1*	Medulloblastoma	c.584 +2T>G	I	[42]
*RAD51C*	Medulloblastoma	c.904+5G>T	I	[15]
*SUFU*	Medulloblastoma	c.1022 +1 G>A	I	[43]
*SUFU*	Medulloblastoma	c. 1365+2T>A	I	[44]
*SUFU*	Medulloblastoma	c.182+3A>T	I	[45]
*SUFU*	Medulloblastoma	c.318-10delT	IV	[45]
*SUFU*	Medulloblastoma	c.1297-1G>C	II	[45]
*SUFU*	Medulloblastoma	c.183-1G>T	II	[46]
*SUFU*	Medulloblastoma	c.684-2A>G	II	[15]
*SUFU*	Medulloblastoma	c.455-1G>A	II	[15]
*TP53*	Medulloblastoma	c.376-2A>G	II	[47]
*WRN*	Medulloblastoma	c.3139-1G>C	II	[15]
*WT1*	Medulloblastoma	c.769+1G>C	I	[15]
*XPC*	Medulloblastoma	c.2251-1G>C	II	[15]
*CHEK2*	Astrocytoma	c.444+1G>A	I	[48]
*NF1*	Pilocytic astrocytoma	c.205_205insTC	III	[49]
*NF1*	Pilocytic astrocytoma	c.1185+1G>A	I	[49]
*NF1*	Pilocytic astrocytoma	c.889-2A>G	II	[49]
*NF1*	Optic pathway glioma	c.2325+1G>A	I	[50]
*NF1*	Optic pathway glioma	c.1260+1G>T	I	[49]
*NF1*	Low-grade glioma	c.6641+1G>A	I	[22]
*NF2*	Ependymoma	c.447+1G>A	I	[22]
*ERCC2*	Diffuse astrocytoma	Not available		[51]
*MUTYH*	Highly infiltrative astrocytoma	Not available		[51]
*ATM*	High-grade glioma	c.7630-2A>C	II	[22]
*MUTYH*	High-grade midline glioma	c.892-2A>G	II	[52]
*MSH6*	Glioblastoma	c.(4002-31_4002-8delins24) + (4002-31_4002-8delins24)	IV	[41]
*NF1*	Glioblastoma	c.1641+2T>A	I	[49]
*NF1*	Anaplastic astrocytoma	c.4174-2>AG	II	[49]
*TP53*	Glioblastoma	c.919+1G>A	I	[49]
*SMARCB1*	AT/RT	c.501-2A>G	II	[53]
*DICER 1*	Pinealoblastoma	c.4050+1G>A	I	[54]
*TP53*	Choroid plexus carcinoma	c.560-2A>C	II	[55]

**Table 3 cancers-14-05967-t003:** Germline pathogenic splice variants found in pediatric patients with sarcomas.

Gene	Diagnosis	Variant	Type of Mutation	Reference
*TP53*	Osteosarcoma	c.671+1G>A	I	[101]
*TP53*	Osteosarcoma	c.672+1G>A	I	[103]
*TP53*	Osteosarcoma	c.375+1G>A	I	[104]
*TP53*	Osteosarcoma	c.559+2T>G	I	[105]
*TP53*	Osteosarcoma	c.672+1G>A	I	[101]
*TP53*	Osteosarcoma	c.770T>A	III	[106]
*TP53*	Telangiectatic osteosarcoma	c.672G>A	I	[102]
*TP53*	Osteosarcoma	c.258+1G>T	I	[107]
*RECQL4*	Osteosarcoma	g.2746del11	IV	[108]
*RECQL4*	Osteosarcoma	g.3685G>A	I	[108]
*RECQL4*	Osteosarcoma	g.2626G>A	II	[108]
*RECQL4*	Osteosarcoma	g.3712del24	IV	[108]
*RECQL4*	Osteosarcoma	c.1391-1G>A	II	[109]
*RECQL4*	Osteosarcoma	c.1704-1G>A	II	[110]
*RECQL4*	Osteosarcoma	c.2059-1G>C	II	[111]
*RB1*	Osteosarcoma	c.940-1G>A	II	[22]
*RB1*	Osteosarcoma	c.2106+2_2106+5del	I	[107]
*NTHL1*	Ewing sarcoma	c.116-1G>A	II	[107]
*SLX4*	Ewing sarcoma	c.1684-1G>A	II	[107]
*FANCA*	Ewing sarcoma	c.523-1G>C	II	[107]
*FANCA*	Ewing sarcoma	c.3828+1G>C	I	[107]
*RAD51C*	Ewing sarcoma	c.905-3_906del	II	[107]
*RAD51C*	Ewing sarcoma	c.1026+5_1026+7del	I	[107]
*CHEK2*	Ewing sarcoma	c.812+1G>T	I	[107]
*FANCC*	Ewing sarcoma	c.456+4A>T	I	[107]
*EXT2*	Ewing sarcoma	c.69+2insAGGG	I	[19]
*FANCD2*	Ewing sarcoma	c.2715+1G>A	I	[19]
*TP53*	Rhabdomyosarcoma	c.560-1G>A	II	[112]
*TP53*	Rhabdomyosarcoma	c.376-1G>A	II	[113]
*TP53*	Embryonal rhabdomyosarcoma	c.783-2A>G	II	[113]
*TP53*	Spindle cell rhabdomyosarcoma	c.560-1G>C	II	[114]
*NF1*	Embryonal rhabdomyosarcoma	c.6704+1G>T	I	[114]
*DICER1*	Embryonal rhabdomyosarcoma	c.1907+1G>A	I	[115]

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
