# Peer review of "Splicing-Disrupting Mutations in Inherited Predisposition to Solid Pediatric Cancer"

_cancers, 2022, doi:10.3390/cancers14235967_

Round 1

Reviewer 1 Report

This is a very timely review highlighting the splicing defect origins of certain pediatric cancers. The importance of this review is further increased due to the authors own finding that there are currently 0 clinical trials ongoing that address splicing-related pediatric cancers.

I have only a single suggestion. It would be good if based on all the 10-100s of incidences of splicing defects in cancers could be summarized in 3-4 subtypes in the form of a figure. Essentially, indicating the different aspects (intronic splicing enhancer, splice site) of splicing that are found to be affected by these mutations. The authors report the mutations but it is often hard to visualize these without the reader having to go and search the genome themselves. Further, like in table II they could indicate what type of effect this mutation has on splicing by calling an entry as Type I, Type II etc. based on their summarized classification from the figure.

This would really boost the impact of this study.

Reviewer 2 Report

In this review by Piedad and colleagues, the authors discuss about the idea that a portion of pediatric cancer have a germline mutation in a cancer predisposition gene. Most of these mutations are found to affect alternative splicing, an important process involved in tumorigenesis. Authors review the importance of mutations in pediatric solid tumor that disrupt splicing and examine the potential of targeted therapies for splicing aberration in cancer. To raise the significance and impact of the work, the authors need to address the following concerns below.

Comments:

- The authors focused mainly on two subgroups of Medulloblastoma (WNT and SHH). However they did not discuss a major paper on the field (Suzuki et al., 2019) where they discovered that SHH patients harbor mutation in U1snRNA. This study explains why SHH tumors have splicing-disruption and aberrantly regulate tumor-suppressor gene PTCH1 and oncogenes GLI2 and CCND2. Please consider discussing those findings. Can they discuss about splicing differences among Medulloblastoma subgroups? WNT, SHH and Group3/Group4 likely arise at different region and time during development. Can you please discuss the importance of splicing during normal development?

- Ependymoma: Authors discuss lightly about the link between epigenetic modification and splicing disruption. PFA Ependymomas express high levels of EZHIP and have a very low H3K27 methylation profile. There are increasing evidences that this phenomenon could be a retained phenotype participating in early tumorigenesis events.

- Ewing sarcoma: There are some papers describing that EWSR1-FLI1 regulates the splicing of Ewing sarcoma and interacts with chromatin remodeling complex.

More importantly, a recent study shows that Ewing sarcomas (and more generally cancers driven by fusion TFs) aberrantly express intergenic region (neogenes) as source of neoantigens. Please consider discussing these findings.

- Therapeutic: Authors mention that there is no approved ASO therapy in cancer. However, several clinical trials using ASO are ongoing. This should be discussed in this review, notably for DIPG and/or HGG. The authors illustrate this therapeutic strategy by focusing on targeting targeting Mnk2b exon 14 by an ASO for Glioblastoma cells. This strategy is promising but authors should considerate to discuss limitation, especially the fact that ASO are often larger than small molecules, thus discussing the importance of the brain blood barrier is necessary.

How many trials targeting splicing disruption are ongoing for pediatric cancer (all type of pediatric cancer), and more generally for adult cancer?
